# Learning-Augmented Robotic Automation for Real-World Manufacturing

*Abstract*— **Industrial robots are widely used in manufacturing, yet most manipulation still depends on fixed waypoint scripts that are brittle to environmental changes. Learning-based control offers a more adaptive alternative, but it remains unclear whether such methods, still mostly confined to laboratory demonstrations, can sustain hours of reliable operation, deliver consistent quality, and behave safely around people on a live production line. Here we present Learning-Augmented Robotic Automation, a hybrid system that integrates learned task controllers and a neural 3D safety monitor into conventional industrial workflows. We deployed the system on an electric-motor production line to automate deformable cable insertion and soldering under real manufacturing constraints, a step previously performed manually by human workers (Video). With less than 20 min of real-world data per task, the system operated continuously for 5 h 10 min, producing 108 motors without physical fencing and achieving a 99.4% pass rate on product-level quality-control tests. It maintained near-human takt time while reducing variability in solder-joint quality and cycle time. These results establish a practical pathway for extending industrial automation with learning-based methods.**

## I. INTRODUCTION

Industrial robots are widely used in manufacturing [1], [2], [3], but most systems rely on waypoint-based programming, which limits adaptability to part variation and environmental uncertainty [4], [5]. While recent advances in learning-based control offer a promising alternative by enabling perception-driven manipulation, their applicability to real-world industrial systems remains challenging due to strict requirements on reliability, cycle time, safety, and limited data availability [6], [7], [8], [9].

In this work, we present Learning-Augmented Robotic Automation, a factory-validated hybrid system that integrates reliable conventional automation with learning-based control in a safety-aware architecture. Instead of replacing existing systems with a single end-to-end policy, our approach retains the core industrial backbone—an explicit task scheduler and pre-taught motions for structured parts of the workflow— while introducing learning only where adaptability is required. This design preserves the predictability, precision, and robustness of classical control in structured sub-tasks while enabling perception-driven adaptation where it is most needed.

We conducted a factory-floor validation (Figure 1). We integrated the proposed system within an existing automation cell on an electric-motor production line and evaluated it through extended long-run operation ($\approx$ 5 h 10 min). The process includes picking motors from random poses, inserting deformable cables into through-holes, and performing soldering. Cable insertion is especially challenging for rule-

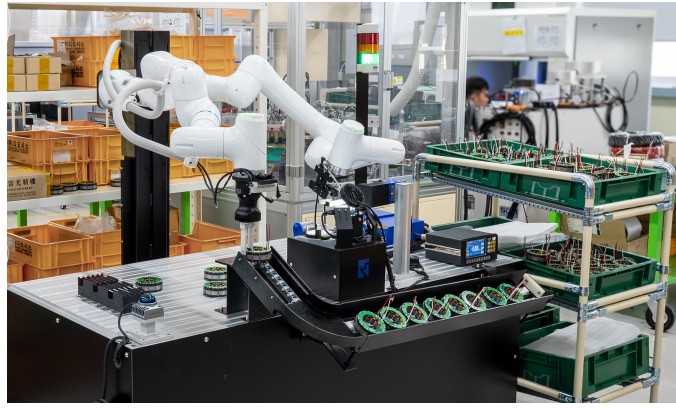

**Fig. 1:** Workstation deployed on the production line

based automation due to material compliance and tight tolerances of less than 0.6 mm. Over 108 consecutive motors with real components and consumables, the system achieved an average cycle time of 159 s (typical human takt time $\approx$ 141 s) and a 99.4% success rate based on product-level Quality Control (QC) tests. This performance was achieved with less than 20 min of real-world data per task.

These results demonstrate the real-world viability of modular learning-augmented automation and establish a practical route to extending industrial automation to tasks that remain difficult for conventional methods.

## II. LEARNING-AUGMENTED ROBOTIC AUTOMATION

Our system integrates learning at two specific levels while retaining a conventional industrial automation backbone: (i) task-level for perception-driven manipulation, and (ii) safety-level for reactive speed and separation monitoring (Figure 2). The remainder of this section details the design, training, and deployment of these learned components.

### A. Learning-Based Task Controllers

We employ two classes of learned controllers (Figure 3a, 3b)—visual servoing and imitation learning—each tailored to different task requirements.

***Visual Servoing***: For kinematics-dominant tasks where achieving the final end-effector pose is more critical than the specific trajectory to reach it, we employ a visual servoing approach (Figure 3a). The visual servoing controller takes the current wrist-camera image $i_t^{wrist}$ as input and predicts a relative end-effector motion $a_{vs} \in SE(3)$, yielding an estimated target pose $s'_f \in SE(3)$ computed as $s'_f = s_t a_{vs}$, where $s_t \in SE(3)$ is the current robot pose, toward which the robot is driven.

*a) Task:* We apply the visual servoing method to the motor grasping task (Figure 2-i). Given multiple motors randomly placed on a table with varying positions and orientations, the robot should grasp a single motor at a time in a desired configuration, where the three PCB holes face the wrist camera with a near consistent orientation (Figure 3a-ii). This standardized grasp reduces variability in hole positions after pickup and simplifies downstream cable insertion and soldering.

*b) Preparing Observation:* To scale the controller to scenes containing multiple motors, the controller operates on masked RGB images, where all regions except the target motor to be grasped are blacked out. These masked observations are obtained using a zero-shot mask tracker, where DINO-based [10] key–query matching initializes prompts and SAM2 [11] refines and tracks the mask (Figure 3a-i).

*c) Deployment:* During deployment, the visual servoing controller operates in an iterative closed-loop manner. At the first step or whenever the previously estimated target pose is reached, the robot is commanded to move toward the newly predicted target pose. This process repeats until convergence, defined by the action norm falling below predefined thresholds ($\| \Delta pos \| < 0.5cm$, $\| \Delta rot \| < 0.5deg$). The iterative refinement is necessary to compensate for partial visibility of PCB holes, visual artifacts from reflections and lighting, and residual regression errors in training. Once the policy converges, visual servoing terminates and the robot executes a predefined grasp by moving down to a fixed table height and closing the gripper.

*d) Training:* To train the neural network policy, we collect data through teleoperation (Figure 3a-ii). For each episode, the motor is initially placed on the table with a random position and orientation. The robot is then manually positioned in a target grasp pose that is ready for motor pickup. Beginning from the target pose $s_0$, the robot is tele-operated via a 3D SpaceMouse to randomly move around the motor, generating a range of perturbed states around the goal pose. The action label is then defined as the relative transformation $a_{vs} = s_t^{-1} s_0$ resulting training pairs $(i_t^{wrist}, a_{vs})$ that supervise the policy to infer corrective motions from visual observations. This tailored data collection strategy, in which the teleoperated trajectories and the recorded action labels are decoupled, is highly data-efficient for kinematics-dominant tasks avoiding the need to collect full reaching trajectories for every motor configuration.

We adopt Action Chunking with Transformers (ACT) [6] as the policy backbone. The action chunk length is set to one, as the policy outputs a single relative target pose per step.

***Imitation Learning:*** For tasks that require high precision or contact-sensitive interaction in semi-structured settings, we employ an imitation learning approach (Figure 3b). Unlike kinematics-dominant tasks that can be handled by sparsely commanding target poses, these tasks require high-frequency adaptive motion based on continuous visual feedback. The imitation learning controller takes multi-camera images and robot state as input and predicts a short-horizon

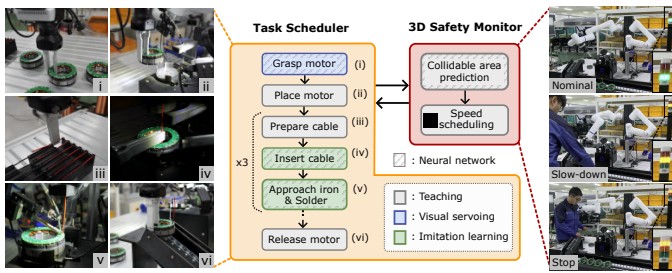

**Fig. 2:** Learning-augmented robotic automation. The software stack comprises modular task controllers and a safety monitoring module. Learned components are integrated at the task and safety levels to overcome the limitation of conventional automation.

trajectory of relative end-effector motions $a_{il} \in SE(3)^K$, often referred to as an action chunk ($K$: chunk size). For modular integration, the policy is also designed to estimate the task success probability.

*a) Task:* We apply the imitation learning formulation to the cable insertion and soldering task (Figure 2-iv,v). These tasks are performed three times for each motor because the motor is three-phase, containing three PCB holes (i.e., hole1, hole2, hole3) and three cables (i.e., red, black, and white). Although the motor grasping controller significantly reduces variability in hole positions after placement on the jig, residual positional uncertainty remains that the above two controllers must handle.

*b) Preparing Observation:* Although raw RGB images are commonly used for imitation learning controllers [6], [12], [7], [13], [9], we empirically found that a structured observation space is particularly beneficial for achieving high performance with limited training data. We design a hole-invariant observation space by considering the characteristics of our task, where the robot performs conceptually the same operation at three PCB holes that appear visually different in raw RGB images.

We first train a lightweight U-Net–based [14] hole mask predictor on camera stream data with semi-automatic labeling via SAM2 (Figure 3b-i). Given a target hole selected by the task scheduler and its predicted mask, we compute the hole's centroid pixel coordinates by averaging the coordinates of all the mask pixels. We also extract a local image crop of size 60×60 pixels centered at the hole. The resulting observation space consists of cropped images around the hole for each camera, the corresponding holes' pixel coordinates, and the robot's relative end-effector position.

*c) Deployment:* During deployment, the imitation learning controller operates until the predicted success probability exceeds a threshold, at which point the task is considered complete. When the cable becomes stuck during insertion due to slight alignment errors, which is often difficult to identify from low-resolution cropped images, we rely on load readings from a load cell mounted beneath the motor jig. When excessive load is detected, the robot moves upward by a random offset and retries insertion.

*d) Training:* The neural network policy is trained using demonstration data collected through teleoperation (Figure

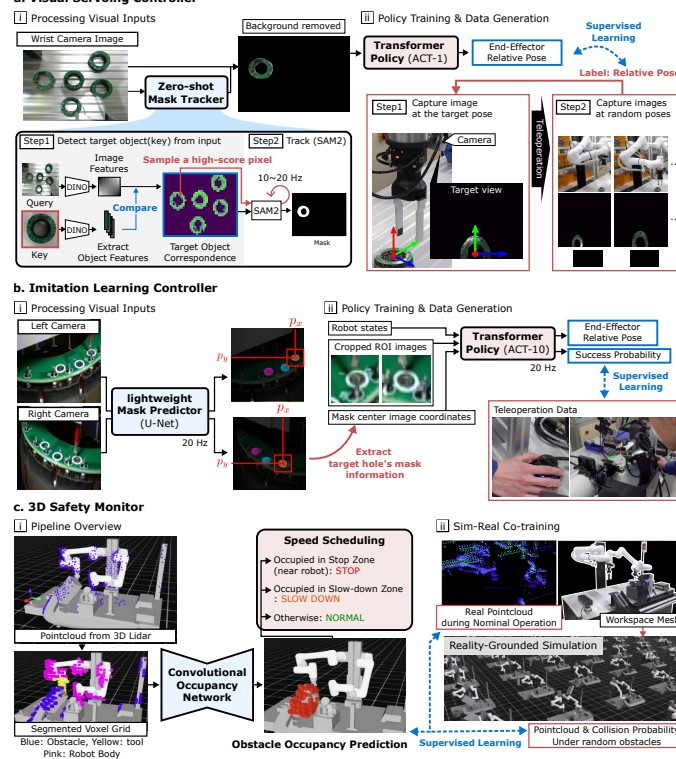

**Fig. 3:** Overview of the proposed method

3b-ii). For each episode, the motor is randomly initialized on the jig within the approximate range of positional variability introduced by the motor grasping controller. The robot is then teleoperated using a 3D SpaceMouse to perform the task, either inserting the cable or approaching the soldering iron tip. During teleoperation, we record images from the left and right cameras mounted on the solder table, the robot's current end-effector pose, and the control inputs from the teleoperation device. The recorded data are processed into training pairs $(i_t^{left}, i_t^{right}, p_t^{rel}, a_{il})$ to supervise the policy, where $p_t^{rel} \in \mathbb{R}^3$ denotes the end-effector position relative to the initial pose of the episode. Binary success labels are generated automatically from demonstrations, with the final portion of each trajectory labeled as successful.

The imitation learning policy uses the ACT model as its backbone. Hole image coordinates are treated as additional state inputs and concatenated with robot proprioception before being passed through a transformer [15] encoder.

### B. Learning-Based 3D Safety Monitor

In conventional automation, safe human–robot interaction is typically ensured by designing systems to comply with Speed and Separation Monitoring (SSM) guidelines [16]. These implementations commonly regulate robot motion using distance measurements from 2D laser scanners [17], [18], [19]. While effective in structured environments, such distance-based strategies are often overly conservative and difficult to deploy in tight human–shared workspaces, where occlusions and complex surrounding structures degrade measurement reliability. To address these limitations, we adopt a learning-based strategy that predicts collidable regions

directly from raw 3D point clouds (Figure 3c). The predicted collidable areas are then used to modulate robot speed between nominal, reduced, and protective stop modes based on predefined safety zones.

***Collidable Area Prediction***: To avoid reliance on explicit online geometric pipelines—such as point matching [20], mapping [21], [22], or handcrafted occlusion handling [23]—we formulate collidable area prediction with a single end-to-end neural network that maps point clouds to occupancy estimates.

*a) Pipeline:* Given a raw point cloud from the 3D LiDAR, we generate a voxelized representation in which each voxel is labeled as empty, obstacle, robot, or tool. The segmented voxel representation is then passed to a neural network to predict collidable areas. We model this network as a Convolutional Occupancy Network [24].

*b) Training:* The neural network is trained using a combination of simulation data and real-world data. We employ a reality-grounded simulation in which each environment is constructed by spawning a 3D mesh of the real-world workspace obtained from CAD models. Robot joint configurations and motions in each environment are sampled around trajectories logged during real-world execution of the automation process to ensure realistic state distributions. Following our previous work [25], each environment additionally includes randomly sized cuboids with randomized motion to simulate external obstacles, such as human workers, that are not present in the nominal setup. Raw 3D point clouds are generated by simulating a 3D LiDAR sensor with randomized mounting perturbations and additive gaussian noise. The neural network is trained using a binary cross-entropy loss with ground-truth occupancy labels for external obstacles generated in simulation.

Due to unmodeled elements such as grasped motors, wired cables, robot tubing, and sensor latency, a model trained solely on simulation exhibits limited sim-to-real transfer and tends to overestimate occupancy in regions without obstacles. To mitigate this issue, we additionally collect real-world LiDAR data while running the automation system without human presence. Since these real-world data do not contain external obstacles, they are labeled as zero occupancy. The network is then co-trained using both simulation and real-world data, with each training batch composed of an equal proportion of samples from the two sources. This co-training strategy allows the network to learn from simulation data to handle occluded point clouds and external obstacles, while adapting to the unmodeled factors using real-world data.

## III. RESULTS

### A. Real-World Production-Line Validation

We validated the proposed learning-augmented automation through continuous operation on a live electric-motor production line at the Neuromeka Pohang factory (Figure 1). The deployment served as a stress test under production-line constraints, including sustained operation, safety for nearby human workers, and downstream product-level QC requirements.

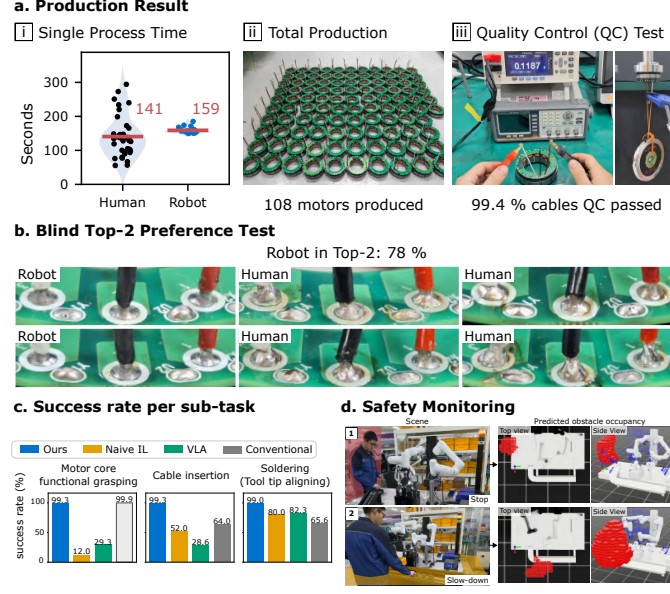

**a. Production Result**

ⅰ Single Process Time    ⅱ Total Production    ⅲ Quality Control (QC) Test

141    159

108 motors produced    99.4 % cables QC passed

**b. Blind Top-2 Preference Test**

Robot in Top-2: 78 %

Robot    Human    Human

Robot    Human    Human

**c. Success rate per sub-task**      **d. Safety Monitoring**

Ours   Naive IL   VLA   Conventional

**Fig. 4:** Production line validation and analysis

Over approximately 5 h 10 min, the system produced 108 motors while maintaining stable operation despite part variability (Figure 4a-ii). This was achieved with less than 20 minutes of real-world data per task; about 8 minutes for motor grasping, 20 minutes for cable insertion, and 4 minutes for soldering.

The robots operated safely in a shared workspace without physical fencing, autonomously slowing down or pausing when workers approached (Figure 4d).

The average nominal cycle time, which excludes safety pauses and slowdowns, was 159 s per motor, which is approximately 12.8 % slower than the average human takt time of 141 s per unit (Figure 4a-i). However, the robot exhibited lower cycle-time variance than human operation, as human workers occasionally incur delays (e.g., exceeding 3 min) when correcting insertion and soldering errors or due to operator fatigue.

Out of 324 cable insertion-and-solder operations, only two failures occurred due to premature triggering of the insertion success detector, leaving the cable insufficiently seated. This corresponds to a 99.4 % success rate. All produced motors passed standard downstream QC tests (Figure 4a-iii).

A blind Top-2 preference test was further conducted with 50 participants on six randomly selected motors (two robot-soldered and four human-soldered) to assess solder-joint quality (Figure 4b). Participants selected both robot-soldered samples in 56% of trials and chose at least one in all cases, with robot-soldered joints receiving 78% of total votes. These results suggest that the robotic system produced solder joints that were visually competitive with—and more consistent than—those produced by human workers.

### B. Comparison with Task Controller Variants

We quantitatively compared our proposed task controllers against representative learning-based and conventional baselines.

The evaluated baselines are as follows:

- **Naive IL** uses the same Action Chunking with Transformers (ACT)-based imitation learning framework as our method [6], but removes our image-processing pipeline and structured visual features.
- **Vision Language Action model (VLA)** employs a model with higher capacity and a stronger visual representation learned from large-scale pretraining. We fine-tuned $\pi_{0.5}$ model [9] for the corresponding tasks with full-resolution RGB images as input.
- **Conventional** follows a traditional robotic automation pipeline based on explicit 3D perception and rule-based waypoints. It estimates 3D poses of the cable tip, the target PCB hole, and the soldering iron tip, and then executes predefined motions based on the estimates.

Each method was individually evaluated on the same hardware, with 150 trials per task (Figure 4c). For cable insertion and soldering, trials were evenly distributed across three PCB holes (50 per hole). For cable insertion, five different cables were used per hole to introduce variation.

Our method achieved over 99 % success across all tasks, outperforming all baselines. Conventional method was assumed near-perfect in motor grasping, as it is being reliably solved using precise 3D sensing and calibration [26].

For the remaining tasks, the conventional baseline was limited by its dependence on explicit 3D geometry: for cable insertion and soldering it achieved approximately 65 % success, limited by noisy depth sensing and imperfect pose estimation. While higher-precision industrial 3D vision solutions could mitigate this issue, they are typically costly.

Other learning-based baselines, including Naive IL and VLA, performed worse than both our method and the conventional baseline on insertion under the same data budget. Despite stronger visual representations or larger model capacity, these methods fail to meet the required task-specific precision in a data-limited setting.

## IV. CONCLUSION

This work investigated whether recent learning-based approaches in manipulation can translate from laboratory demonstrations to reliable operation in real manufacturing. Our factory deployment results showed that structured integration of learned components can deliver stable long-run operation under production-line constraints.

Taken together, these results illustrate a practical pathway toward more flexible and adaptive automation. Although we evaluated a soldering task, the design principles—modular task decomposition, task-specific inductive biases in both observation and action, and learned safety-aware coordination—are applicable to other tasks with geometric variability, difficult-to-handcraft behaviors, and imperfect fixturing. The results show the potential of learned modules to expand the automation frontier and unlock new classes of automatable tasks.

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
