# OpenReview forum: "Learning-Augmented Robotic Automation for Real-World Manufacturing"
_IEEE.org/ICRA/2026/Workshop/Manipulation_Robustness — ICRA 2026_

### Official Review · Reviewer_T5iJ · 2026-05-02

**Rating:** 7
**Confidence:** 4

**Review:**

This paper presents a real-world manufacturing study, and its main contribution lies in the combination of a modular learning-based method with convincing long-horizon deployment results. The method is thoughtfully designed, using different controllers for grasping versus insertion/soldering and integrating a learned 3D safety monitor into a conventional automation backbone, which makes the overall system both technically sound and practically relevant. The experiments include production-line validation, baseline comparisons, cycle-time analysis, QC outcomes, and a human preference study on solder quality.

Strengths: This method do not force a single end-to-end policy, but instead introduce task-specific inductive biases and structured observations that are well matched to the requirements of each manipulation sub-task. The experimental validation, which goes beyond typical robotics demonstrations by showing sustained operation in a live factory setting with meaningful measures of robustness, quality, and consistency.

Weaknesses: The method remains heavily engineered, relying on task decomposition, custom perception pipelines, success detection, and recovery heuristics, so its scalability to new tasks is still unclear.

---

### Decision · Program_Chairs · 2026-05-21

Accept